



# Brief communication: On the environmental impacts of 2023 flood in Emilia-Romagna (Italy)

Chiara Arrighi[1], Alessio Domeneghetti[2]

[1]Department of Civil and Environmental Engineering, University of Firenze, Florence, 50100, Italy
5 [2]Department of Civil, Chemical, Environmental and Materials Engineering, Alma Mater Studiorum Università di Bologna, Bologna, 40136, Italy

*Correspondence to*: Chiara Arrighi (chiara.arrighi@unifi.it)

**Abstract.** The impacts of floods to environmental assets are often not assessed. This communication reflects on this issue by analysing the reported environmental consequences of the 2023 Emilia-Romagna flood. The information on the 10 environmental impacts is constructed by collecting data from reports, press releases and interviews in the aftermath of the event. The most frequently reported damages involve water resources and water-related ecosystems, with cultural and supporting ecosystem services particularly affected. Indirect effects in time and space, intrinsic recovery capacity, cascade impacts on socio-economic systems and lack of established monitoring activities appear as the most challenging aspects for future research.

## 15  1 Introduction

Floods are among the most damaging natural hazards (Munich Re, 2023). Direct economic damages such as those to homeowners or companies are well known since affected citizens ask for compensation to insurance companies or public authorities. On the other hand, damages to public properties, such as natural areas, fade to the background because they claim less attention than other losses in the media and in the political agenda. Moreover, for a long time, the belief that, 20 being a natural phenomenon, flooding has limited effect on natural ecosystems has probably prevented a deep investigation on its potential impacts on the environment. However, we are not anymore in the time when flooding brought relevant benefits for human activities (e.g., Nile River floods for agriculture in ancient Egypt). Nowadays natural areas have been shrinked by human development and urbanizations and reduced to spots of territory (in some cases even protected by law to avoid their disappearance) immersed in anthropized areas (Ren et al., 2023). Human activities have reduced floodplains 25 worldwide (Rajib et al., 2023), also reducing the recurrence of flood events and altering riparian ecosystems (Walker et al., 2022). Moreover, when a flood occurs, many sources of pollution might be affected by flood waters and undesired substances can be transported to many different environmental matrices, phenomena referred as natech events (Suarez-Paba et al., 2019).

Flooding can have a significant impact on the biodiversity of natural ecosystems and the degree of impact depends on 30 species, environmental and flood characteristics (e.g., scour of river bed sediment) (Zhang et al., 2021; Francoeur and Biggs,



2006). Flooding also influences the abundance of riverine species, their migration patterns, and the ingress of non-native species (Sueyoshi et al., 2023; Thomaz, 2022).

Although ecologists have been studying specific problems related to the influence of flooding on specific habitat or species, systematic mapping of environmental areas or habitats potentially exposed to flooding is often not included in flood risk
management plans (e.g., maps and risk management plans required by the EU Flood Directive). This could also be attributed to the fact that several characteristics of the environmental damages are still not clear or are difficult to be qualitative and quantitative estimated. Perhaps, the aspects deserving more attention is, first, the identification of the types of natural environments that can be threatened or damaged by floods, second, the quantification of the potential degree of damage, its nature (indirect, direct, intangible, monetizable) and its persistence in time with respect to the natural resilience of
ecosystems. Finally, it is also worth mentioning that the potential for cascade effects to human health and economy is still mostly unrevealed.

This brief communication aims at bringing up the subject of flood impacts to environment by highlighting the impacts reported after the 2023 flood in Emilia-Romagna on natural environment and resources. Specifically, the objective is to reflect on the lack of methodological background and shared methods for the territorial analysis of flood induced
environmental impacts by identifying: (1) available data and their representativeness, (2) the most significant environmental consequences and affected ecosystems, (3) the physical mechanisms behind such consequences. A greater consciousness on the relevance of these topics is essential to foster additional investigations and larger attention by our community.

## 2 Case study and method

### 2.1 The 2023 Flood in Emilia-Romagna (Italy)

In May 2023 the Emilia-Romagna (ER) region (Northern Italy), one of the most economically developed area in the country, experienced severe and spatially extended flooding. This was the consequence of particular hydrological and weather conditions reached after two major intense rainfall events that hit the same area: a first event on May 1-4 and a second one on May 16-18. The second event, in particular, was due to a vast area of low atmospheric pressure affecting the entire central Mediterranean basin that channelled moisture-laden air masses from North African coasts toward the Italian peninsula. The
cyclon circulation moved the air masses towards central Italy, where the combination with cold air coming from North and hills of the Appennines caused the persistence of heavy rainfall over Eastern ER and Marche regions. Several gauging stations on hill basins recorded rainfall amount larger than 200 mm in 48 hours (the higher value ever recorded in some cases since their installation), with an overall estimation of roughly 350 million of cubic meters of water (equivalent to that on average expected on a 6-month period over the same region) fell in 36 hours (Barnes et al., 2023).

The impact of such event was devastating, also because hitting basins and rivers already tried by intense rainfall (up to 200 mm in some locations in two days) and floods (i.e., levee overtopping and failures) occurred after the May 1-4 event: 23 rivers overflowed affecting 100 municipalities; more than 400 landslides severely damaged infrastructures (e.g., road,



railways, electrical networks, etc.) also limiting rescue activities. In total, the event caused more than 36000 displaced and 15 deaths (Agenzia per la sicurezza territoriale e la protezione civile, 2023).

The area affected is one of Italy's most important economic region, relevant for agriculture and industrial activities, as well as for its touristic attraction. Thus, a first rough estimate pointed out a damage of 8.8 billion euros, a value that probably depicts a ratio of the real economic impact.

Actually, the region is also characterized by relevant and important natural resources (e.g., the UNESCO biosphere reserve of the Po River delta) that were affected by the flooding too. However, impacts (economic and not) on them are hardly
estimated and not considered in the current economic damage estimates.

## 2.2 Data collection and analysis

Differently from other type of damages (i.e., private houses, economic activities, etc.), environmental impacts of the event were not systematically recorded by Authorities. In the aftermath of the Emilia-Romagna flood, from June to September
2023, a web-based search of the string "Romagna flood environmental damage" ("*alluvione Romagna danni ambientali*" in Italian) has been carried out and repeated in time to look for newspaper articles, press releases, interviews, and videos. Whenever possible the areas where environmental damages were reported have been georeferenced. Flooded areas have been retrieved from the Copernicus rapid mapping service and other tool available to local Authorities (e.g., ESA mapping services). All the observed impacts have been listed, described, and then grouped according to the physical cause of the
damage and the driving flood mechanism responsible for the damage. The type of environmental assets exposed, and the main ecosystem services affected have been reconstructed based on the search of environmentally protected areas listed in National and Regional geographic databases.

## 3 Results and discussion

### 3.1 Observed environmental impacts

Figure 1 shows the map of inundated areas (blue shades), protected areas (green dashed line) and the reported impacts containing enough information for their georeferencing. Protected areas include Natura 2000 areas, wetlands, national and regional parks. According to the collected data, 12 different types of environmental impacts have been observed, which are summarized in Table 1. The most frequent impact observed is the bathing ban issued by the Regional Environmental Protection Agency (ARPAE, 2023) in 11 beaches due to levels of faecal microorganisms above bathing water quality limits
(light blue symbol in Fig. 1). In fact, the flood event has affected wastewater treatment plants (WWTPs) and sewerage systems with a consequent release in rivers of untreated wastewaters.





The second most frequent environmental impact reported is the interruption of nesting of species, such as seagulls, terns, avocets, flamingos, and black-winged stilts (magenta symbol in Fig. 1) (Costa, M., 2023). This impact occurred in coastal wetlands and river mouths that were flooded, particularly in three areas listed according to Ramsar convention, i.e., the salt

pan of Cervia (National Natural Reserve), *Ortazzo e Ortazzino* and *Pialassa della Baiona* (both Natura 2000 areas, in the Regional Park of the Po River Delta).

**Figure 1: Setting of the study area (top right), map of inundation, protected areas and environmental impacts observed.**

Other reported impacts were algal blooms and death of fishes in rivers (some of them without precise geographic position) that were reported as consequences of high concentration of nutrients and anoxy in freshwaters due to WWTPs and sewerage inefficiencies and wash-off from agricultural surfaces (ARPAE, 2023). Reports of damages to plants (halophyte) and fauna (e.g., bivalves, micro invertebrates) were also described for several species typical of hypersaline ecosystems, altered by freshwater ingress (Costa, M., 2023; Bovenzi, M., 2023). Also, the sudden invasion of alien species (*Callinectes sapidus*) in

these environments has been attributed to change in salinity (Costa, T., 2023).





Finally, one reported case of contaminated soil affected the area of the municipality of Conselice where floodwaters stagnated for about two weeks and residents were evacuated for sanitary reasons. Other impacts mentioned involved riparian habitats of flooded rivers due to erosion of banks and vegetation removal (Protezione Civile Emilia-Romagna, 2023).

As summarized in Table 1 (last column) three main flood related processes are here recognized as responsible of damage to environmental assets namely: (i) contamination and transport of pollutants, (ii) submersion, (iii) erosion.

**Table 1: Summary of impacts and their proposed classification. The * symbol identifies additional impacts without associated geographic information (not reported in Figure 1).**

| Observed impact | Description | Cause | Classification of flood processes generating environmental impacts |
|---|---|---|---|
| Bathing ban | *escherichia coli* in sea water above safety limits for bathing water quality | Persistent flood, sewerage and WWTP failure | Contamination and transport of pollutants |
| Soil contamination | Deposition/stagnation of sediments and wastes in flooded area | | |
| Algal bloom | Change of colour of surface river water (red or dark) | High nutrients concentration and consequent anoxy | |
| Death of fishes | Fishes found dead in channels and rivers | | |
| Water contamination* | Presence of hydrocarbons and other pollutants (e.g., fertilizers) in surface waters and flood waters | Release of fuels by tanks and wash-off of agricultural surface | |
| Interruption of nesting | Nests of several bird species have been submerged or swept away during reproductive season | Submerged nesting area | Floodwater submersion |
| Damage to bivalves | Bivalves found dead in coastal lagoons where shellfish farming is carried out | Hypersalin ecosystem alteration | |
| Damage to fishes and invertebrates | Death of species and alteration of reproductive season | | |
| Damage to halophyte plants | Plants and flora withering due to freshwater submersion | | |
| Invasion of alien species | Ingress of alien species (*Callinectes sapidus*) and increased competition with native species due to changes in coastal lagoon salinity | | |
| Damage to riverbanks | Destruction of animal shelters and interruption | | Erosion |





| habitats* | of nesting in riverbanks | Bank erosion/ levee failure | |
|---|---|---|---|
| Damage to riparian vegetation* | Interruption of nesting | | |

## 3.2 Space dimension of impacts

From a spatial point of view, the map of Fig. 1 clearly shows how reported environmental impacts of the flood occurred downstream of the most inundated areas, i.e., river mouths, coast and coastal wetlands. The identification and characterization of these impacts is however rather accidental and cannot be seen as the outcome of an established monitoring activity of the environmental damages due to floods. This major event occurred right before the start of the bathing season that, according to law, entails an increase in frequency of coliform monitoring activity performed by public

authorities. Such circumstances enabled spotting the impact even in areas not directly affected by inundation. On the other hand, the attention on bathing areas might have led to an underestimation of water quality impacts along upstream rivers stretches, where monitoring was not performed, despite being the origin of the pollution. Some exception apart, as in cases where the Environmental Agency was called in emergency due to sudden reported red colour of river waters, impacts on natural rivers were not considered and thus, monitored. Similarly, spring is the reproductive season of many bird species,

which brought the attention to the impacts of the event on coastal ecosystems of ecological value (Fig. 1 green shaded areas). These two aspects have contributed to push the attention on the reported impacts towards the coast.

## 3.3 Time dimension of impacts

From a temporal perspective it is possible to recognize both a direct time-dependent exposure, e.g., nesting activity occurs in a specific season, and an indirect flood damage which occurs and persists after the event.

In fact, the bathing bans disappeared with different velocities according to the location and after about 30-45 days from the flooding all bathing waters were reported safe. About the interruption of nesting, expert reported that a second nesting attempt this year was improbable for most of the bird species, due to a late occurrence of the flood with respect to reproductive time. No information was available in relation to the persistence of polluted soils and freshwaters anoxic conditions or to the recovery of flora and fauna in wetlands and riparian habitats.

## 135 3.4 Impact metrics

According to the collected information, the most affected environmental assets have been the water resources and water ecosystems, i.e., freshwater, transitional and marine waters, and terrestrial ecosystems, including Natura 2000 areas of biodiversity conservation and Ramsar wetlands. It should be noticed also that these areas were neither reported as flooded, i.e., exposed, as permanent water bodies are automatically excluded by flood maps during satellite image processing.



In terms of losses to ecosystem services the most affected categories have been the recreational functions (cultural values) in case of the prolonged bathing ban, and biodiversity functions (supporting value) when animal and plant species have been affected. Valuing flood losses in terms of a reduction of provided ecosystem services could set the base for a quantitative damage metric for environmental assets. Moreover, although river water quality was not systematically monitored, we can imagine losses also to regulating and provisioning services since the importance of ecological status of water bodies in

providing such ecosystem services has been demonstrated (Grizzetti et al., 2019).

Damages to environmental assets extend their consequences to economic activities, such as tourism and food production (e.g., aquafarming), thus although the mentioned losses are mostly intangible, some cascading aspects could be monetizable. Repercussions of environmental damage to human health are still unrevealed, although the increase of diseases, such as hepatitis or gastrointestinal disorders due to ingestion of contaminated water, could be potentially monitored.

Most of the observed damages, as well as the natural recovery capacity of environmental systems are, however, difficult to quantify with the present technical and scientific knowledge.

## 4 Conclusions

In this brief communication the ex-post analysis of the 2023 Emilia-Romagna flood highlighted the occurrence of different types of environmental impacts especially on water bodies (rivers, bathing seawaters, and coastal wetlands in particular),

water and terrestrial ecosystems, with supporting and cultural ecosystems services affected by the event. The three main flood processes identified as responsible for damages to environmental assets are (i) contamination and transport of pollutants, (ii) submersion, (iii) erosion.

However, flood damages to environmental assets are often overlooked. This happens since damages to ecosystems are hardly monetizable and quantifiable, and because of the lack of established monitoring activities devoted to their assessment.

Moreover, most natural ecosystems have their inner capability to self-recover after a shock, without human intervention (i.e., costs), unless tipping points are overcome. Environmental impacts show different time and space patterns, also due to seasonal drivers. Though, such dynamics remain largely unknown due to the paucity of information (even in scientific literature) and absence of specific monitoring protocols.

From the analysis of this event some general reflections arise with regards to our consciousness of the environmental impacts

of floods, of their extent and current practices for their monitoring. With respect to other types of impacts, e.g., structural losses to a bridge, flood damages to environment require a more holistic and multi-disciplinary understanding able to embrace the underlying ecological system dependencies among temporal and spatial phenomena. From a temporal perspective, environmental losses can occur immediately through direct contact with water, e.g., submersion of nests, but might have prolonged indirect repercussions, e.g., on later population dynamics. In this sense, the resilience of each

environmental matrix (i.e., soil, water) or even each living species (e.g., plants, birds) has its own dynamics and natural recovery ability, which requires specific expertise. From a spatial point of view, environmental flood impacts might also
extend beyond the directly inundated area, e.g., coliforms in bathing waters downstream of flooded areas, thus revealing their intrinsic direct and indirect nature. The flood impacts to several ecosystems services also suggests cascade consequences on health and socio-economic activities, such as water abstraction, aquaculture, tourism, etc., which again remain largely unclear and difficult to quantify.

Further research should focus on better understanding what are the environmental areas that can suffer from floods and their specific vulnerabilities. Moreover, future research should move towards the conceptualization of potential direct, indirect and cascading impacts of flooding on the environment to support their inclusion in flood risk mapping and management processes.

**Conflict of interest**

The authors declare that the research was conducted in the absence of any commercial or financial relationships that could be construed as a potential conflict of interest.

**Data Availability**

The georeferenced impact data will be available upon request.

**Author contribution**

Conceptualization, data curation, analysis, visualization, writing-original draft, and funding acquisition: CA.

Data curation, writing-original draft, writing-review and editing: AD.

**Funding**

This study was carried out within the RETURN Extended Partnership and received funding from the European Union Next-GenerationEU (National Recovery and Resilience Plan – NRRP, Mission 4, Component 2, Investment 1.3 – D.D. 1243 2/8/2022, PE0000005)

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
