# Peer review of "Brief communication: On the environmental impacts of 2023 flood in Emilia-Romagna (Italy)"

_Natural Hazards and Earth System Sciences, 2023_

## Referee Comment (RC1)

**nhess-2023-184-- Brief communication: On the environmental impacts of 2023 flood in Emilia-Romagna (Italy)**

This manuscript presents an in-depth analysis of the environmental impacts resulting from the 2023 flood in Emilia-Romagna. It highlights the often overlooked damages to natural ecosystems caused by flooding and emphasizes the necessity for a more in-depth understanding of these impacts. The overall quality of this manuscript is good, my comments are the following:

1. The manuscript emphasizes the difficulty in quantifying environmental impacts of flood. However, presenting a framework or proposing potential metrics, such as resilience-based measures, to assess environmental impacts and the recovery process could enhance the study's practical applicability.

2. Although this manuscript is focused on the 2023 flood in Emilia-Romagna, a comparative analysis with similar flood events or a discussion on global trends in this domain could enrich the discussion.

3. Further exploration of potential implications for policy-making and flood risk management strategies to prepare for and reduce the adverse impacts of flood to natural environment might be useful.

4. The inclusion of a deeper exploration of soil erosion and sedimentation in various environmental settings affected by the flood could enhance the manuscript's comprehensiveness.

5. There are a few instances of typo, a thorough proofreading would enhance the overall readability of the manuscript.

---

## Author Response (AR1)

**Reply to referee 1**

We would like to thank the anonymous referee for the comments, which are clear and pertinent. The present understanding of flood impacts of environment is still at its first research stages, therefore we can only partly address the raised comments, although recognizing that the mentioned aspects, i.e., resilience-based metrics, analysis of other flood events, and policy implication, will require further research. A point-by-point reply is provided below, and the manuscript will be modified accordingly to incorporate these reflections while preserving its brief-communication nature.

1. The manuscript emphasizes the difficulty in quantifying environmental impacts of flood. However, presenting a framework or proposing potential metrics, such as resilience-based measures, to assess environmental impacts and the recovery process could enhance the study's practical applicability.

    Reply. The identification of metrics for flood impact is very useful to describe the losses, however as highlighted in the communication, the heterogeneity of damage sources and potential consequences, their spatial and temporal dimensions make very difficult to select specific metrics. The main issue is the dynamic behaviour of environmental matrices and their intrinsic resilience, therefore the time at which quantifying the impacts can be different for each ecosystem. Proposing resilience-based metrics is appropriate, but after a deeper understanding of the behaviours of different ecosystem compartments, which are still poorly known and also depend on the damaging sources. **We incorporated this reply at LL. 140-145 and LL.162-164 of the revised manuscript.**

2. Although this manuscript is focused on the 2023 flood in Emilia-Romagna, a comparative analysis with similar flood events or a discussion on global trends in this domain could enrich the discussion.

    Reply. From a national perspective, we analysed also the Marche region flood occurred on September 2022, however, due to the period of occurrence (end of the bathing season) no assessment of the presence of coliforms in bathing waters was available. Moreover, no significant environmentally protected areas were flooded except for the Parco Nazionale dello Zolfo di Marche e Romagna (National Park of Sulphur) where some riverine ecosystems were affected due to bank erosion and consequent human restoration. In particular, the habitat of river crab was seriously compromised (M. Fraternale, personal communication) but no official data from news or public authorities were available.
    At a European level, the flood of July 2021 raised the attention of the media on chemical pollution due to flooding (Völker et al., 2023), on acute and unusual release of organic pollutants due to river sediment remobilization (Weber et al., 2023; Schwanen et al., 2023), which ultimately might affect human health. In particular, also in the review by Völker et al. the release of untreated wastewaters from affected facilities was highlighted. Moreover, the authors found that media stated that environmental and health risks were poorly discussed. In our work we highlight that also from a scientific point of view these topics are understudied.
    Remobilization of pollutants in river sediments during floods are probably the most frequently reported environmental effect also in other flood events. In the work by Weber et al. (2023) the scale of analysis was a single city and the focus on the contamination of deposited sediments (with metals and organic compounds) with no description of potential effects on flora and fauna. In the work by Schwanen et al., (2023) the scale of analysis was the Rur catchment and the focus again on flood sediments deposited in the floodplain. They again recognized that sediments can accumulate contaminants which are mobilized by the floods but without considering environmental or ecological effects.
    Besides these works on 2021 European flood, most of the attention was focused on direct losses (e.g. to infrastructures) (Jonkman et al., 2023). This confirms that there is a lack of interest or policy prescription at EU level to better understand environmental consequences of floods. **These new comments and references have been added to the manuscript at LL.110-121 of the revised manuscript.**

3.  Further exploration of potential implications for policy-making and flood risk management strategies to prepare for and reduce the adverse impacts of flood to natural environment might be useful.

    Reply. We believe that a first attempt to drive change in policy is a better scientific understanding of potential consequences of floods on natural and environmental areas. In fact, reducing adverse consequences of flood on natural environments is not straightforward. For instance, reducing the occurrence of flooding on a wetland might compromise its hydrological functioning. Therefore, we believe that environmentally protected areas should be included in flood risk management plans with a more holistic description of their behaviour. However, including natural environments on flood risk assessment is not straightforward, since this indirectly entails a better knowledge and estimate of their values (under different perspective: ecologic, social and economic, etc.), which require additional investigations too.

4.  The inclusion of a deeper exploration of soil erosion and sedimentation in various environmental settings affected by the flood could enhance the manuscript's comprehensiveness.

    Reply. The issue of soil erosion and sedimentation is critical for resuspension of contaminants (see reply to point 2) but it can be critical also for the alteration of ecology of the first layers of soil. Unfortunately, this aspect has not been reported by the media or public authority and it can be additionally mentioned to highlight a further gap of knowledge. This topic, together with others risen during our surveys on the recent flood events, further stresses the multidimensionality of such issues, which call for multidisciplinary efforts to reach a comprehensive assessment of the environment impacts.

5.  There are a few instances of typo, a thorough proofreading would enhance the overall readability of the manuscript.

    Typos have been checked.

**New references added**

Jonkman, S. N., Moel, H. De, Moll, J. R., and Slager, K.: Editorial for the Special issue on " 2021 Summer Floods in Europe " Setting the scene : an unprecedented event The Netherlands : Post-flood fact finding studies, 2, 1–7, 2023.

Schwanen, C. A., Müller, J., Schulte, P., and Schwarzbauer, J.: Distribution, remobilization and accumulation of organic contaminants by flood events in a meso-scaled catchment system, Environ. Sci. Eur., 35, https://doi.org/10.1186/s12302-023-00717-4, 2023.

Völker, C., Friedrich, T., Kleespies, M. W., Marg, O., and Schiwy, S.: "The toxic substance has killed all ducks": framing of chemical risks related to the 2021 summer flood in German news media, Environ. Sci. Eur., 35, https://doi.org/10.1186/s12302-023-00789-2, 2023.

Weber, A., Wolf, S., Becker, N., Märker-Neuhaus, L., Bellanova, P., Brüll, C., Hollert, H., Klopries, E. M., Schüttrumpf, H., and Lehmkuhl, F.: The risk may not be limited to flooding: polluted flood sediments pose a human health threat to the unaware public, Environ. Sci. Eur., 35, https://doi.org/10.1186/s12302-023-00765-w, 2023.

**Reply to referee 2**

The paper "Brief communication: On the environmental impacts of 2023 flood in Emilia-Romagna (Italy)" analyses the environmental consequences of the 2023 Emilia-Romagna flood.

The importance of this paper is in the impact event analysis carried out just in the aftermath of the event, preventing the loss of information usually occurring after events occurrence, especially concerning environmental impacts that are so difficult to detect an analyze. Besides, the paper also represents a methodological path of data collection for future events.

The paper is well written and all the references concern recent researches, denoting the attention the authors put in preparing the manuscript.

I can give some suggestions to further improve the paper:

1) Figure 1: I suggest to eliminate the descriptions for each point on the map. Actually, the points are classified in different colors, then you can explain the meaning of the colors in the legend (that can be placed under Italy in the gray area) and you can eliminate texts from map.

2) table 1: I suggest to improve formatting style. You can enlarge the II and III columns in order to avoid two-lines cells, and you can decrease the interline in the entire table.

3) LINE 165: "With respect to other types of impacts, e.g., structural losses to a bridge, flood damages to environment require a more holistic and multi-disciplinary understanding able to embrace the underlying ecological system dependencies among temporal and spatial phenomena". This sentence is unclear: structural losses to a bridge, in my opinion, are a kind of impact outside the topic of the paper. I suggest to modify or eliminate.

General Reply

We would like to thank the referee #2 for understanding the challenge of detecting and analysing flood impacts to environment. We took care of improving the readability of the Map in the revised manuscript. We tried to adjust the Table but we could not improve as much as expected, we believe that this aspect will be further addressed in the typesetting phase.
Moreover, we better clarified the discussion in the suggested sentence. **LL.179-181 of the revised manuscript.**